# Role of MicroRNA in Inflammatory Bowel Disease: Clinical Evidence and the Development of Preclinical Animal Models

**DOI:** 10.3390/cells10092204

**Published:** 2021-08-26

**Authors:** Kanika Suri, Jason A. Bubier, Michael V. Wiles, Leonard D. Shultz, Mansoor M. Amiji, Vishnu Hosur

**Affiliations:** 1Department of Bioengineering, Northeastern University, Boston, MA 02115, USA; kanika.suri@takeda.com; 2The Jackson Laboratory, Bar Harbor, ME 04609, USA; jason.bubier@jax.org (J.A.B.); michael.wiles@jax.org (M.V.W.); lenny.shultz@jax.org (L.D.S.); 3Department of Pharmaceutical Sciences, Northeastern University, Boston, MA 02115, USA; m.amiji@northeastern.edu

**Keywords:** miRNA, IBD, ulcerative colitis (UC), Crohn’s disease (CD), genetic diversity, collaborative cross, diversity outbred, humanized mice, extracellular vesicles

## Abstract

The dysregulation of microRNA (miRNA) is implicated in cancer, inflammation, cardiovascular disorders, drug resistance, and aging. While most researchers study miRNA’s role as a biomarker, for example, to distinguish between various sub-forms or stages of a given disease of interest, research is also ongoing to utilize these small nucleic acids as therapeutics. An example of a common pleiotropic disease that could benefit from miRNA-based therapeutics is inflammatory bowel disease (IBD), which is characterized by chronic inflammation of the small and large intestines. Due to complex interactions between multiple factors in the etiology of IBD, development of therapies that effectively maintain remission for this disease is a significant challenge. In this review, we discuss the role of dysregulated miRNA expression in the context of clinical ulcerative colitis (UC) and Crohn’s disease (CD)—the two main forms of IBD—and the various preclinical mouse models of IBD utilized to validate the therapeutic potential of targeting these miRNA. Additionally, we highlight advances in the development of genetically engineered animal models that recapitulate clinical miRNA expression and provide powerful preclinical models to assess the diagnostic and therapeutic promise of miRNA in IBD.

## 1. Introduction

Inflammatory bowel disease (IBD) is a multifactorial progressive disease marked by recurrent chronic inflammation of the gastrointestinal tract, a dysregulated immune system, and dysbiosis [1,2,3,4]. IBD comprises two main forms—Crohn’s disease (CD) [5] and ulcerative colitis (UC) [6,7]. Abdominal pain, chronic diarrhea, weight loss, exhaustion, and anorexia are common symptoms among younger CD patients. Bloody diarrhea is a symptom of chronic CD. Although the entire gastrointestinal tract can be impacted in CD patients, the terminal ileum and colon are the most affected. Preferred therapeutic agents for CD include immunosuppressants (e.g., azathioprine, mercaptopurine, and methotrexate), corticosteroids, anti-TNF therapy (e.g., Adalimumab and Infliximab), monoclonal antibodies (e.g., Vedolizumab and Ustekinumab), and surgery for patients who do not respond to treatment [8,9]. In UC, only the colonic mucosa is inflamed, and the major symptoms include rectal tenesmus, bleeding, diarrhea, abdominal pain, and fecal incontinence. Preferred therapeutic agents include, 5-Aminosalicylic acid (oral, suppository, or enema), corticosteroids, anti-TNF agents (e.g., Infliximab, Golimumab, Adalimumab), and monoclonal antibodies (e.g., Vedolizumab and Ustekinumab) [8,10]. In the United States, approximately 3 million adults (>18 years of age) have IBD [11], and there is currently no cure. The global prevalence of IBD is increasing [12,13], necessitating the development of innovative therapeutic strategies. The mechanisms underpinning IBD pathogenesis are still emerging; nevertheless, a large body of evidence suggests that microRNAs play a major role in IBD pathophysiology [14,15,16]. Here, we review clinically relevant miRNAs that have been validated in various mouse models of IBD.

## 2. MicroRNA and IBD

Non-coding RNAs (ncRNA) are a subset of cellular RNAs that do not code for protein, but ncRNAs play vital roles in regulating a multitude of cellular processes. According to the Encyclopedia of DNA Elements (ENCODE) project and subsequent studies, approximately 20% of the human genome is not transcribed [17]. 

Based on function, ncRNAs are divided into housekeeping and regulatory ncRNAs. Housekeeping ncRNAs include ribosomal RNA (rRNA), transfer RNA (tRNA), small nuclear RNA (snRNA), small nucleolar RNA (snoRNA), telomerase RNA/tRNA-derived fragments (TERC tRF), and tRNA halves (tiRNA). Housekeeping ncRNAs are expressed in all cell types and are essential for cell viability through supporting RNA translation into proteins and transcript splicing [18]. Regulatory ncRNAs are sub-classified on the basis of size into small ncRNAs (sncRNAs), which are usually less than 200 nucleotides (nts) in length, and long ncRNA (lncRNAs) greater than 200 nts [19]. sncRNAs include microRNA (miRNA), small interfering RNA (siRNA), and piwi-interacting RNA (piRNA) [17]. Some ncRNAs, such as enhancer RNA (eRNA), promoter-associated transcripts (PATs), and circular RNAs (circRNAs), exhibit variable lengths and can hence be classified as either sncRNAs or lncRNAs [17]. Regulatory ncRNAs play critical roles in modulating and maintaining transcriptional and translational processes [20]. 

### 2.1. Synthesis and Physiological Significance of miRNA

The most widely studied ncRNA is miRNA. miRNA is 18–25 nts long with a primary function in regulating post-transcriptional gene silencing by binding to the 3′-untranslated regions (UTR) of messenger RNA (mRNA), thereby blocking translation. It is estimated that over 60% of human coding genes are modulated by miRNAs at a translational level, with about two-thirds of all protein-coding genes under pressure to maintain miRNA pairing [21]. miRNA is promiscuous; a single miRNA typically regulates the translation of multiple mRNAs. On the other hand, multiple different miRNAs may be required to block the translation of a particular mRNA [22,23]. 

miRNA originates in the nucleus from the introns or exons of DNA, transcribed by RNA polymerase II and III into a 1 kb long primary-miRNA (pri-miRNA) that contains a hairpin-like structure [24]. Drosha and DGCR8 cleave the pri-miRNA into precursor-miRNA (pre-RNA) approximately 70 nts long. The pre-miRNA is then transported out of the nucleus with the help of Exportin 5. In the cytoplasm, the RNA-induced silencing complex (RISC), which contains double-stranded RNA binding proteins including Dicer and trans-activator RNA binding protein, cleaves the pre-miRNA into a short, 18–25 nt long, double-stranded mature miRNA. The Argonaut 2 (Ago2) protein present in the RISC holds on to the 5′ end of the miRNA strand, called the guide strand—the target strand for the 3′ UTR of the mRNA. Next, RISC finds the target mRNA strand complementary to the guide miRNA strand and, subsequently, either degrades or inhibits the translation of the mRNA by binding to the ‘seed region’ spanning 2–8 bases from the 5′ end of the miRNA [25]. Within the cytoplasm, miRNA functions are kept in check and regulated by interactions with other ncRNAs. For example, the circRNA hsa_circ-0001368 binds to miR-6506-5p, acting as a miRNA sponge that inhibits the miRNA’s suppression of the forkhead family transcription factor FOXO3, thus functioning as a tumor suppressor in gastric cancer [26]. As noted above, interactions between miRNA and mRNA lead to the blocking of translation into protein. For example, miRNAs miR-122, -192, -495, and -671 bind to the many 3′-UTRs of mRNA coding for nucleotide-binding oligomerization domain-containing protein 2 (NOD2), also known as caspase recruitment domain-containing protein 15 or inflammatory bowel disease protein 1 (IBD1). This binding represses NOD2 expression in response to inflammatory stimulation by muramyl dipeptide (MDP) in HCT116 cells [27]. These complex interactions of miRNA underscore its importance in determining cell fate, enabling cells to navigate through their developmental stages and maintaining a cell at its appropriate developmental stage [28,29].

### 2.2. Detection of miRNA

Ambros et al. have set out some identifiers for distinguishing miRNA from siRNA. For miRNA specifically, two out of the following criteria should be met: (i) expression criterion: the expression should be confirmed by tests, such as northern blot, or quantitative real-time polymerase chain reaction (RT-qPCR); (ii) structure criterion: one arm of the hairpin precursor must contain the miRNA structure; (iii) conservation criterion: phylogenetical conservation of sequences [30,31]. Since miRNA is being increasingly studied to understand its functions in the normal state and as a disease biomarker, the improved detection of miRNA is expected to aid in prognosis and in identifying disease-relevant, putative therapeutic miRNA candidates. What makes miRNA detection complicated is the short sequence length of the mature miRNA, its low abundance, and tracing a miRNA to a specific cell or tissue type [32]. Commonly used detection methods include Northern blot, RT-qPCR, and high-throughput methods, such as microarrays and NanoString. The choice of detection method depends on the purpose of the experiment. For example, when looking for miRNA signatures of disease, high-throughput methods are used to first define the repertoire of miRNAs implicated in the disease state, and then a small subset of these are validated by RT-qPCR [33,34,35].

### 2.3. Therapeutic Potential of miRNA

Various approaches support the leveraging of miRNA for therapeutic intervention, depending on its role in a given disease. To silence a miRNA upregulated in disease, antisense oligonucleotide “antagomirs”, or their chemically derived versions, such as locked nucleic acids (LNA), as well as miRNA sponges, including circRNA, can be used. Chemically modified oligonucleotides with higher affinity to the miRNA ‘seed region’ can be used to block the miRNA’s effect and to also potentially interfere with the miRNA-induced silencing complex [36]. miRNA sponges, which are longer nucleotides, can prevent miRNA from binding to its mRNA targets [37]. Still, the utility of sponges as therapeutics is limited given their potential off-target effects on exogenous nucleic acids. In diseases characterized by miRNA deficiency (leading to the overexpression of an mRNA), miRNA replacement therapy can be implemented [38] using mimics or “agomirs”. Combination therapies have also shown promise. For instance, in oncology, miRNA mimics have been used synergistically with anti-cancer treatments to render tumor cells that are more susceptible to chemotherapy or radiotherapy [39].

miRNA mimics/agomirs and antagomirs need to be encapsulated to retain their function. Lipid carriers (e.g., liposomes, lipid nanoparticles) and polymeric carriers (e.g., cationic carriers such as polyethyleneimine) along with viral vectors, conjugates, and exosomes can be explored for the delivery of miRNA therapeutics [38]. Decorating the surface of lipid and polymeric nanoparticles with specific molecules can help to improve targeting to desired cell and tissue types.

### 2.4. Limitations in Current Methodologies in miRNA Research

Given the ability of a single miRNA to regulate multiple genes, targeting miRNA has the potential to simultaneously impact several disease processes, thus making miRNA a more rewarding modality over siRNA, which is mRNA-specific. The downside of this, however, is potential off-target effects, as well as undesired on-target effects, highlighting the necessity of better understanding miRNA-regulated gene networks and critical nodes therein. To advance miRNA-based therapeutics, the field needs to address challenges in three main areas: [31,38]: (i) miRNA-regulated genes and gene networks; (ii) efficient miRNA delivery; (iii) animal models that recapitulate critical aspects of IBD and enable testing the physiological role of miRNA and the impact of miRNA-targeted interventions. 

miRNA and its putative networks remain incompletely understood. Until we have a more comprehensive view of the pathways affected by miRNAs, there will be a high burden to validate both on-target and off-target effects. Bioinformatics databases, such as TargetScan [40] and miRBase [41] are available to predict putative miRNA targets to help understand the affected networks. While these databases are a great resource for finding potential mRNA targets, the experimental validation of the binding of miRNA(s) to target mRNA(s) is a necessary validation step. Standardization in miRNA identification and in tissue sourcing and sampling (e.g., fresh frozen tissues vs. formalin-fixed embedded tissues) must also be clarified to ensure data reproducibility. Furthermore, since human and animal tissue samples are often heterogeneous, determining the precise cellular origin of miRNA is challenging.

While simpler model organisms, most notably nematodes, have been utilized to discover miRNAs and to identify their molecular functions, more complex and physiologically relevant models are sought to study the role of miRNA in human disease. Since there are higher risks of off-target effects with miRNA therapies, preclinical validation in animal model systems is imperative to progress miRNA as a therapeutic modality into the clinic. While there is substantial research on miRNA mimics’ and antagomirs’ molecular stability and affinity, which will aid in making miRNA therapy more efficacious [38], selection of a suitable predictive animal model to study and test miRNA mechanisms, and interventions is an equally important part of the preclinical process.

Given the pleiotropic nature of miRNAs, finding candidate miRNAs dysregulated in a disease state could help develop miRNA-based therapies with the potential of correcting complex networks of pathogenically disrupted proteins. This review article will explore the mouse models used to study IBD, miRNAs implicated in these models and in IBD patients, as well as potential nanoparticle miRNA therapeutics for IBD (Figure 1).

## 3. Developing Novel Mouse Models of IBD

IBD is a multifactorial disease with four major inter-related contributors to its etiology—genetics, environment, dysbiosis, and immune system abnormalities [42]. While the exact etiology is unknown, these four main factors affect specific phenotypes that are characteristic of IBD—immunopathogenic changes, alterations in resident gut microbes, loss of epithelial barrier function, and polarization of T-cells [43]. Various IBD animal models recapitulate some of these standard features, and the choice of animal model is dependent on the desired aspect of the disease to be studied. Broadly, IBD mice models can be divided into four groups, based on how the disease is induced: (1) chemically induced models, (2) spontaneous colitis models, (3) gene knockout (KO) models, and (4) T cell transfer model of colitis [44]. The most commonly used IBD mouse models are listed in Table 1. Importantly, these models can be combined with humanization methodologies (transgenes and cell transplantation) to generate state-of-the-art IBD models.

### 3.1. Humanized Mouse Models

In recent years, numerous therapeutic candidates have failed to succeed in clinical trials at least in part because researchers have used experimental mouse models that do not sufficiently recapitulate the human immune microenvironment. IBD is an autoimmune disease involving complex interactions between multiple innate and adaptive immune cell populations. Additionally, genetic factors, environmental triggers, and the gut microbiome play significant roles in IBD. The differences between the mouse and human immune systems—a sophisticated network of cells, tissues, and organs that has uniquely evolved to defend against invading microorganisms—can thus fundamentally impact the fidelity of a mouse model in mimicking disease pathogenesis. Thus, we may need to examine human autoimmune diseases in the context of a functional human immune system in preclinical animal models. The ideal model would allow for the testing of disease pathogenesis, longitudinally enable detailed study of the human immune system during the disease course and be economically feasible. For instance, severely immunodeficient mice homozygous for targeted mutations in the IL-2 receptor γ-chain locus (*IL2rg*), particularly NOD-*scid* *IL2rg^null^* (NSG) mice, support heightened levels of engraftment with human hematopoietic stem cells (HSC) and, notably, increased human T cell development. These mice lack host T cells, B cells, and NK cells, facilitating more accurate in vivo study and the modeling of human autoimmune diseases. The development of humanized mice has been recently reviewed elsewhere [62,63,64]. 

Although functional human immune systems develop following human HSC engraftment of NSG mice, human T cell function is not optimal. Human T cells in engrafted NSG mice are educated on the mouse major histocompatibility complex (MHC). Hence, Shultz and colleagues have developed new stocks of NSG mice that lack murine (MHC) class I and II and transgenically express human MHC HLA class I and class II molecules to enhance human T cell function following engraftment with HSC from human donors [65,66,67,68]. These mice can be used to study and test regulatory T cell (Treg) therapies for IBD. 

To develop a humanized mouse model of colitis, Goettel et al. [69] engrafted mice selectively expressing an HLA class II allele with human CD4^+^ T cells and induced colitis in reconstituted humanized mice using 2,4,6 Trinitrobenzenesulfonic acid (TNBS). TNBS treatment (a single rectal enema containing 0.25 mg of TNBS in 50% ethanol) of NSG mice selectively expressing human HLA-DR1 in the absence of mouse MHC class II (NSG Ab^o^ DR1), engrafted with or without human CD4^+^ T cells from an HLA-DR1 donor, resulted in significant weight loss only in NSG Ab^o^ DR1 mice engrafted with human CD4^+^ T, but not in the non-reconstituted mice. Furthermore, using histopathological analysis, the authors demonstrated extensive infiltration of CD4^+^ T cells into the colonic lamina propria and crypt, as well as goblet cell loss with edema, fibrosis, and transmural inflammation in NSG Ab^o^ DR1 engrafted with human CD4^+^ T cells, but not in non-reconstituted NSG Ab^o^ DR1 mice. Lastly, using qPCR, the authors showed increased transcript levels of *TNF*, *IFNG*, *IL2*, *IL4*, and *IL17A*. Together, these data suggest that in the humanized TNBS-treated mice, human CD4^+^ T cells mediate disease pathology, and more importantly, these mice enable the testing of human therapeutics targeting CD4^+^ T cells to treat colitis. 

Treg stimulation is a promising therapeutic method for improving immunological tolerance. In the humanized mouse model of colitis, Goettel et al. [69] used ITE [(2-(1′ H-indole-3′-carbonyl)-thiazole-4-carboxylic acid methyl ester], a ligand of the cytosolic aryl hydrocarbon receptor (AhR) transcription factor, to activate Tregs and suppress effector T cells. The authors observed that ITE treatment increased the number of Tregs while decreasing disease severity. Furthermore, in human T cells, ITE therapy boosted transcript levels of multiple transcription factors associated with anti-inflammatory pathways. These data further demonstrate the utility of the humanized TNBS-treated mice in evaluating the role and targetability of T cell subsets.

Humanized mice have also been used in attempts to stratify patients for treatment with specific therapies. Recently, Jodeleit et al. [70] engrafted NSG mice with peripheral blood mononuclear cells (PBMC) from 40 patients with ulcerative colitis (UC) following sensitization of the mice with intrarectal application of 10% ethanol and tested the response of these mice to Adalimumab therapy (Humira; FDA-approved for UC and CD). Analysis of the immunological profile of the PBMC donor patients revealed two major subgroups that reflected the dynamics of inflammation. The NSG recipient mice reconstituted with the donor PBMCs and treated with Adalimumab mirrored the inflammatory phenotype of the patients.

### 3.2. Genetic Diversity and IBD

Most mechanistic biomedical research in mice has been conducted using a single inbred strain, C57BL/6. Studying disease in one strain of mice is equivalent to performing research on one single person. Thus, it is not surprising when the results do not translate across diverse patient populations. This limited genetic background may be one of the contributing factors to the replication crisis [71]. IBD mice models can be divided into four groups based on how the disease is induced in mice. During the past two decades, genetic researchers have combined these different approaches to identify the genetic contribution to disease. Studies have taken advantage of classic inbred mouse strains, such as C3H/HeJ and C57BL/6J, to identify modifier loci, QTL, that affect the severity of the IL-10 deficient mouse pathology [72,73]. A major finding of this work was identifying a distal chromosome 3 locus related to induced colitis susceptibility and other QTL on Chr 1, 2, 4, 5, 8, 12, 17, and 18. Recent studies have utilized the diverse genetics of the *Mus musculus* subspecies strain PWD/PhJ to identify regions on Chr 1 and 2 that contain loci associated with high susceptibility to induced IBD [74]. The importance of genetic diversity to phenotype variation has been discussed extensively (reviewed in [75,76])) and argues in favor of a population-based approach to modeling human disease.

The recently derived mouse population, the diversity outbred (DO), and associated collaborative cross (CC) strains allow for the interrogation of the genetic diversity of 45 million SNPs, similar to the diversity in the human population [77]. These strains of mice are derived from an intercross of eight classical inbred strains to create a panel of 50–75 recombinant inbred lines (CC) having a genomic architecture derived from the eight founder strains. These strains were outbred to produce the DO mice, which are heterozygous and unique, with each mouse having a unique genetic makeup. Within the panel of CC strains, there are phenotypes that, due to transgressive segregation, are more diverse than any of the original eight founder strains. In producing the CC strains, CC011/UncJ was identified as a strain that develops spontaneous colitis. Utilizing this population to identify modifier loci, four QTL were mapped across the genome that explain 27.7% of the disease severity variation seen in this population [78]. Findings from these modern genetically diverse populations demonstrate the ability to identify susceptibility loci that impact our ability to model human disease.

### 3.3. Importance of Choosing the Right Mouse Model(s) to Study and Validate miRNA as Therapeutic Targets

miRNA plays a significant role in determining immune cell fate decisions and regulating cell signaling pathways [79]. Conversely, changes in the cell signaling pathways bring about differential expression of miRNAs associated with that pathway. When using IBD animal models, different aspects of the complex disease are recapitulated in each model, and dysregulated pathways associated with these aspects will have a corresponding miRNA or set of miRNAs differentially expressed, likely in a tissue-type specific manner. Thus, to maximize translational relevance, it is essential to understand the consequences of dysregulated miRNA expression as well as the cell signaling pathways linked with this dysregulation.

Differential expression of miRNAs dysregulated in various mouse models of IBD highlights the importance of choosing the most appropriate mouse model for the given question of interest. For instance, in a comprehensive study, Wu et al. [51] demonstrated that the expression of miRNAs varies depending on the choice of the colitis mouse model. The authors found that several miRNAs, including miR-21, miR-223, miR-10b, and miR-142-3p, show differential expression patterns between two commonly used mouse models of colitis—dextran sodium sulfate (DSS)- and TNBS-induced colitis (Table 1). Whereas the expressions of miR-21 and miR in the DSS model, they were upregulated in the TNBS-model. Likewise, the express-223 downregulated ion pattern of miR-10b and miR-142-3p were inconsistent between the DSS- and TNBS-models, indicating the inherent differences between these two mouse models of colitis. Additionally, when the authors chose miR-21 for further in vivo analysis, as it is upregulated in the inflamed colonic mucosa of patients with UC [34,35,80] and CD [34,80,81,82,83], they observed that the deletion of miR-21 in mice significantly suppressed DSS-induced, but not TNBS-induced, inflammation and weight loss. While this study clearly demonstrates the significance of miRNAs in the pathogenesis of IBD, and that miRNAs are potential therapeutic targets in IBD, this also indicates the care required in model selection, with ideally more than one mouse model being used for preclinical validation of therapeutic targets. 

## 4. MicroRNA as a Therapeutic Target in IBD

The epithelial barrier in the gut houses the largest volume of immune cells, as it interfaces with the external environment [84]. The gut undergoes several insults through pathogenic, chemical, or biological sources and, in healthy conditions, can overcome and repair without causing chronic inflammation [85]. Several cells of both hematopoietic and non-hematopoietic origin guard the gut to maintain homeostasis [84]. Aberrant activation of the immune system due to various confounding factors can lead to chronic inflammation. Indeed, several miRNAs are associated with the maintenance of mucosal immunity. Thus, it is not surprising that the upregulation of pro-inflammatory cytokines that characterizes chronic inflammation is associated with perturbation of upstream regulatory miRNA networks. Clinical evidence implicates deregulated miRNA expression in serum samples and intestinal tissue biopsies of UC and CD patients compared to healthy controls [33,34,35,80,81,82,83,86]. In this section, we discuss the miRNAs known to be associated with common pathophysiological pathways in IBD, specifically (a) epithelial barrier disruption and (b) immune system dysregulation. We also discuss studies that have evaluated their therapeutic prospects in preclinical models.

(a) Epithelial barrier disruption: The epithelial cells lining the gut interface with the external environment and the gut microbiome on the luminal side. Integrity between epithelial cells allows passive transport of nutrients from the lumen while restricting pathogen entry. Tight junctions (TJ), adherens junctions (AJ), gap junctions, and desmosomes connect adjacent cells, regulate the transport of molecules from the luminal side, and participate in various intracellular signaling pathways [87,88,89]. They form the apical junctional complex (AJC). Selective paracellular permeability based on charge and size is controlled via TJs [90]. The TJ is made up of transmembrane proteins (occludin, tricellulin, claudins, and junctional adhesion molecules) and peripheral membrane proteins (Zona occludens (ZO)-1,2,3 and cingulin) [91]. AJs play a role in cell–cell adhesion, and tissue development and homeostasis [89]. Gap junctions aid in nutrient transfer from one cell to another [89]. Desmosomes impart structural support to the cells in the tissue [89]. Mucosal inflammation brings about several changes in these stable intercellular junctions, thereby perpetuating chronic inflammation in diseases such as IBD. These junctional proteins have hence become lucrative targets to alleviate IBD.

Another class of peptides that play a role in maintaining homeostasis is the Trefoil Family (TFF), which is secreted by mucus-producing cells and goblet cells in the lumen to maintain homeostasis [92]. TFF3 is a type of TFF most commonly found in the colon, has anti-apoptotic properties [93] and has been shown to induce cell migration in vitro [94]. Several miRNAs contribute to epithelial integrity and homeostasis by regulating the expression of proteins involved in the maintenance of epithelial barrier integrity. Below, we discuss the therapeutic potential of targeting these miRNAs.

(a.1) miRNA therapeutic intervention in epithelial barrier disruption (Table 2): *OCLN* encodes occludin, a tight junction protein responsible for regulating paracellular transport. One study analyzed colonic biopsies for occludin mRNA expression from 54 individuals divided into three groups of active UC (*n* = 20), remission UC (*n* = 16), and healthy controls *(n* = 18) [95]. They found that occludin expression in the remission UC group was significantly lower than in the control group, and in active patients was higher [95]. According to TargetScan, several miRNAs, including miR-200b-3p, miR-200c-3p, miR-122a, and miR-429, target OCLN mRNA. For instance, in Caco-2 cells, TNFα, a pro-inflammatory cytokine upregulated in IBD patients, induces the degradation of occludin mRNA through the rapid expression of miR-122a [96]. Treatment with modified antisense miR-122a (dose: 25 nM, complexed with Lipofectamine 2000) reversed this effect when tested in the intestinal perfusion model [97]. There was an increase in occludin mRNA levels, suggesting that antisense miR-122a has the potential to be explored for barrier maintenance [96]. Another pro-inflammatory cytokine that affects occludin expression is IL-1β. IL-1β has been implicated in modulating intestinal permeability and in IBD pathogenesis [98,99,100]. A recent study by Rawat et al. revealed that the intraperitoneal administration of IL-1β in mice caused a marked decrease in occludin mRNA, but an increase in miR-200c-3p levels [101]. To establish the link between occludin mRNA and miR-200c-3p, the authors inhibited miR-200c-3p expression in Caco-2 cells stimulated with IL-1β and observed increased occludin mRNA levels [101]. Additionally, in the DDS colitis model (3%), an oral gavage of antagomiR-200c-3p (dose: 800 mg/day, starting two days before DSS treatment and continued for seven days of DSS course), prevented the loss of occludin expression and maintained the barrier function [101].

In UC patients’ colonic biopsies, lower levels of Occludin and ZO-1 are linked to significantly greater levels of miR-21. Yang et al. transfected Caco-2 cells with miR-21 mimics to determine the underlying molecular mechanisms and found that miR-21 mimics resulted in not only reduced levels of Occludin and ZO-1, but also increased permeability, as evidenced by a decrease in the transepithelial electrical resistance (TEER) across the Caco-2 monolayer [123]. Shi et al. reported that miR-21 KO mice are much less sensitive to DSS-induced colitis (3.5%) than control mice, which they attribute to miR-21-mediated intestinal permeability modulation [102].

Claudins are a family of 27 transmembrane proteins that enable epithelial cell–cell adhesion, establishing paracellular barrier function, among other functions [124]. One such protein, CLDN1, was shown to be upregulated in *Mir-29^−/−^* mice, which show loss of miR29a and miR29b activity [103]. These mice were able to tolerate small intestine and colon barrier disruption better than wild-type mice when treated with TNBS to induce colitis [103]. Target validation studies revealed that miR-29a and b regulated NFKB Repressing Factor (NKRF) mRNA, another barrier protection gene, in addition to CLDN1 [103]. These gene knockout studies suggest that the inhibition of miR-29a and miR-29b is a potential therapeutic strategy to restore intestinal permeability in IBD [103]. Another study elucidated that miR-233 mediates crosstalk between the IL-23 pathway and the intestinal barrier through the regulation of Claudin 8 (CLDN8) expression. IL-23 plays a key role in the pathogenesis of IBD [125,126,127]. Wang et al. [104] tested the hypothesis that IL-23 suppresses CLDN8 expression through miR-223 to mediate IBD, by inhibiting miR-233 expression via the intraperitoneal administration of antagomir-223 (dose: 7.5 mg/kg, prepared as 3 mg/mL in PBS, dosed for three successive days 24 h after TNBS administration) in the TNBS-colitis model. The authors observed that the inhibition of miR-223 restored CLDN8 expression, restored barrier function, and alleviated inflammation, resulting from TNBS-induced colitis [104]. 

Two AJC proteins, NUMB and CLDN11, help maintain intestinal barrier integrity. The AJC can be disrupted by MALAT1 (Metastasis Associated Lung Adenocarcinoma Transcript 1), a lncRNA that is found to be downregulated in the colonic intestinal mucosa of CD patients [105]. A recent study revealed that MALAT1 affects intestinal permeability by sequestering miR-146b-5p, which targets the 3′ UTR of NUMB and CLDN11 mRNA and thus suppresses their expression [105]. In a TNBS model of colitis, treatment with antagomir-146b-5p, but not a negative antagomir control, restored barrier integrity by enhancing the protein levels of NUMB and CLDN11 [105].

Cingulin, expressed in the cytoplasm, interacts with many components of AJs and TJs. Notably, cingulin expression is significantly downregulated in UC patients [128]. A study explored the cingulin-miRNA interaction and found that miR-24 is overexpressed in the crypts of colonic biopsies and the serum of UC patients compared to healthy controls [128]. Further experiments in Caco2 cells to elucidate the association between cingulin and miR-24 in disruption of intestinal barrier integrity revealed that miR-24 targets cingulin mRNA to cause barrier disruption. Thus, it might be beneficial to explore miR-24 inhibitors to enhance cingulin expression and restore barrier function in UC patients. 

Hypoxia has also been studied in the context of IBD, and several studies showed that hypoxia inducible factor 1 subunit alpha (HIF-1α) inhibitors improve intestinal barrier function when administered orally [129] or intraperitoneally [130]. miR-135a was found to be suppressed in both UC and CD patients [86]. Lou et al. [131] showed that the DSS model (4% DSS) recapitulates human colitis and results in the significant downregulation of miR-135a. Furthermore, the authors showed that HIF-1α is a target of miR-135a, and that the downregulation of miR-135a enhances HIF-1α expression to promote apoptosis and inflammation. Consistently, the overexpression of miR-135a in colonic epithelial cells prevented apoptosis and inflammation by inhibiting the expression of apoptotic factor Bax and enhancing the expression of anti-apoptotic factor Bcl-2 via HIF-1α [131]. Another study found a negative correlation between miR-16 and Bcl-2 expression in Caco-2 cells and in the colonic mucosa of DSS-treated mice [107]. Further investigation in 3% DSS induced colitis mice (two weeks) with the intraperitoneal administration of anti-miR-16 (dose: 5 mg/kg, twice per week) alleviated colitis and reduced proinflammatory cytokines in comparison to PBS-treated DSS controls [107].

Along with TJs, which help maintain the integrity of the intestinal epithelial barrier, TFFs help maintain, regenerate and restore barrier function in response to acute injury and chronic inflammation [92]. TFF3, a type of TFF, was shown to be negatively regulated by miR-7 in human colonic epithelial cells LS174T [132,133]. Furthermore, miR-7-5p levels were upregulated in the inflamed lesions of pediatric patients compared to uninflamed tissue [133] and TNBS-treated mice [106]. Subsequently, the team explored the effects of the inhibition of miR-7 in a preclinical colitis model [106]. Tail vein administration of 100 nmol/kg of antagomir-7 2 h after TNBS administration and subsequent assessment of the intestinal tissue harvest seven days after treatment revealed improved histological colitis scores, higher TFF3 production, and a reduction in colitis-induced damage [106].

miRNA targets that are relevant to epithelial permeability in the context of other diseases could potentially also be applied to IBD. For example, miR-155 is upregulated in the intestinal epithelial tissues in severe acute pancreatitis animal models, resulting in the upregulation of TNF-α in the serum [134]. This, in turn, causes the downregulation of various AJC proteins such as ZO-1 and E-cadherin, disrupting the intestinal epithelial barrier [134]. Similarly, in an animal model of intestinal ischemia-reperfusion, miR-21 was upregulated, which correlated with the upregulation of several pro-inflammatory cytokines, and thereby intestinal barrier disruption [135]. Moreover, several miRNAs also play a role in central nervous system disorders due to their role in maintaining the integrity of the blood–brain barrier (reviewed in [136]), and further research is needed to ascertain their possible relevance in IBD. 

(b) Immune system dysregulation in IBD: The large volume of immune cells in the gastrointestinal tract ensures the safekeeping of internal organs. These cells constantly interact with ingested materials while maintaining homeostasis with the gut microbiota [84]. Although these are foreign, the host immune system under healthy conditions is not overstimulated in the presence of microbiota, thus maintaining a delicate immune balance in the gut. The first line of defense is the innate immune system, which, through NOD-like receptors (NLRs) and pattern recognition receptors (PRRs) sense foreign microorganisms and initiate an immune response to dispose of the invaders. PRRs are present on immune cells, such as macrophages and dendritic cells (DCs), epithelial cells, and myofibroblasts, which together maintain gut immune homeostasis [137]. The endpoint effects of PRR stimulation are generally the release of pro-inflammatory cytokines. Macrophages and DCs also act as antigen-presenting cells (APCs), linking innate and adaptive immune systems. The adaptive immune system is the second line of defense and is more nuanced than the innate immune system. Information from APCs helps prime the adaptive immune cells to orchestrate the immune response via T cells and B cells.

In IBD, innate sensing dysregulation causes an aberrant increase in the levels of pro-inflammatory cytokines, which leads to chronic inflammation if not checked and resolved. In addition, during chronic inflammation, the dysregulation of the adaptive immune system sustains the levels of pro-inflammatory cytokines. Inflamed tissues in IBD patients are infiltrated by activated immune cells, which initiate a cascade that ultimately leads to the overexpression of pro-inflammatory cytokines, such as TNFα, IL-6, IL1B, IFN-Y, IL-17A, IL-12, IL-23, IL-18, and the downregulation of anti-inflammatory cytokines, such as TGF-B, IL-10, IL-25, IL-33, IL-37 [138]. Below, we summarize studies indicating how targeting miRNAs alleviates an overactive immune system. 

(b.1) miRNA therapeutic intervention in immune system dysregulation (Table 2): NFκB signaling transcribes pro-inflammatory cytokines and plays a crucial role in innate and adaptive immune responses [139]. Acting as a fine-tuner of innate signaling, miR-146a targets Tumor necrosis factor receptor-associated factor 6 (TRAF6) and Interleukin 1 Receptor Associated Kinase 1 (IRAK1) that are both upstream proteins in the NFκB signaling pathway, keeping the production of pro-inflammatory cytokines in check [140]. A colonic biopsy study has shown that miR-146a is significantly elevated in UC patients compared to CD patients [64]. Another study showed elevated miR-146 expression in both UC and CD pediatric patients, compared to intact mucosa [141]. The elevation could be due to TLR4 activation, which induces the transcription of miR-146a via the NFκB pathway [140]. Notably, miR-146a is not restricted to myeloid cells but is also expressed by intestinal epithelial cells [109,142]. miR-146a KO mice were more susceptible to endotoxin challenge and developed spontaneous autoimmune disorders upon aging [143]. A recent study also elucidated that, by targeting TRAF6 in intestinal epithelial cells (IECs) and Receptor Interacting Serine/Threonine Kinase 2 (RIPK2) within the myeloid cells, miR-146a can keep IL-17 signaling in check, repressing inflammation further and suppressing tumor growth [109]. The same group further demonstrated that treatment, with miR-146a mimics administered intraperitoneally (dosed at 5 mg/kg) in the DSS (3%) colitis model, improved disease outcome [109]. Along with the negative regulation of TLR/IL-1R signaling, miR-146a controls Treg cell conversion into IFN-γ-producing Th1 cells mediated via the signal transducer and activator of transcription 1 (STAT1) [144] and, thus, controls hyperresponsiveness in T cells [145], further underscoring its therapeutic utility in preventing excessive inflammation.

A study explored the inhibition of miRNAs in human colonocytes (NCM460) and its effect on NFκB activation via stimulation by IL-6. From a list of 348 miRNA inhibitors explored, the inhibitor of miR-214 suppressed phospho-NFκB most efficiently (>90%) [110]. The group further found that miR-214 in colonic tissues of UC patients was expressed 8-fold higher, compared to control subjects [110]. Mechanistically, IL-6 stimulation induces the upregulation of STAT3-mediated transcription of miR-214, leading to reduction in the levels of phosphatase and tensin homolog (PTEN) and PDZ and LIM domain 2 (PDLIM2), ultimately increasing the phosphorylation of AKT and activation of NFκB. The higher expression of miR-214 is observed in long-term UC patients (>10 years) compared to short-term UC patients (0–10 years). Furthermore, in mice, DSS-induced colitis significantly upregulates miR-214 expression. Utilizing the DSS model of colitis, the authors examined whether intracolonically administered miR-214 antagomirs (dose: 12 mg/kg, 2.5 mg/mL diluted in PBS, 4 doses, every 2 days after DSS treatment regime) reduce disease severity. Expectedly, miR-214 antagomirs significantly reduced the disease activity index in DSS-treated mice [110]. 

Tian et al. recently studied the effect of inflammation on miR-31 and found that the inflamed mucosa of UC and CD patients had elevated miRNA levels, while the levels were reduced to uninflamed levels in patients under remission [108]. NFκB and STAT3 pathway activation upregulated miR-31 in human colon epithelial cell line LOVO and primary mouse colon organoids. Furthermore, DSS- and TNBS-induced colitis in miR-31 KO mice developed more severe colitis than in WT mice. The group also found that the overexpression of miR-31 improved colitis in TNBS treated mice. Further investigation revealed miR-31′s role in suppressing *IL6st, IL17r, and IL17ra* genes within the colonic epithelial cells, all implicated in IBD. Further target evaluation also revealed the role of miR-31 in promoting epithelial regeneration by regulating the WNT and Hippo signaling pathways. Given the ability to target multiple genes at once, miR-31 mimics were utilized as a therapy by encapsulating the mimics in peptide-based oxidized konjac glucomannan (OKGM)-PS microspheres (3.15 µg) administered via enema. The therapeutic intervention reduced 3.5% DSS-induced colitis severity and recovery in WT mice, both in preventative and therapeutic dosing regimens, respectively, bolstering the utility of both miR-31 mimics and the delivery system [108].

The NLRP3 inflammasome is assembled in response to the innate sensing of microbiota and damage-associated molecular patterns (DAMPs), and the disruption of this process can lead to dysbiosis and pyroptosis within myeloid cells, both of which eventually cascade into chronic inflammation, as in IBD [146]. miR-223 regulates the NLRP3 inflammasome [147] and plays a critical role in myeloid cell development, negatively regulating proliferation, cell fate, and activation [148]. Dysregulated miR-223 expression is observed in IBD, chronic obstructive pulmonary disease [149], rheumatoid arthritis [150], and type 2 diabetes [151,152]. In IBD, miR-223 is upregulated in the colonic mucosa of patients with UC [80,83] and CD [33,80,81,82,83], as well as in animal models of colitis [48,49,51,58]. Neudecker et al. elucidated that the miR-223 controls enteric inflammation by targeting the NLRP3 inflammasome [111]. In mice, DSS upregulates miR-223 levels in the colon as early as day two after treatment. To study the source of miR-223 induction, neutrophils were deleted in *B6/J* mice via antibodies, followed by DSS treatment. Neutrophil depletion significantly reduced miR-223 expression, indicating that infiltrating neutrophils could be the source of miR-223 [111]. Furthermore, since miR-223 targets the 3′ UTR of NLRP3, studies using miR-223 KO mice [148] showed higher NLRP3 protein levels in the colon of DSS-treated mice when compared with untreated controls. Furthermore, blocking NLRP3 significantly improved the disease pathology in DSS-treated miR-223 KO mice [111], indicating a potential therapeutic strategy using miR-223 mimics to target the NLRP3 inflammasome. To this end, the authors retro-orbitally administrated a nanoparticle emulsion encapsulating miR-223 mimics (dose: 50 µg, complexed with NLE: DOPC, squalene oil, PS-20, and an antioxidant) on days 1 and 3 after 3% DSS treatment in wild-type mice. miR-223 significantly mimics suppressed NLRP3 mRNA and attenuated the transcript levels of several pro-inflammatory cytokines, including IL-1β, IL-6, and TNF [111].

Macrophage phenotypes exist in a spectrum between M1, representing a pro-inflammatory-skewed phenotype, and M2, representing an anti-inflammatory-skewed phenotype. The IRF5 transcription factor is known to cause M1 differentiation [153,154]. It is well known that *IL-10* KO mice spontaneously develop colitis [59,155]. Investigating this model for macrophage polarization revealed abundance in the M1 phenotype [112]. Furthermore, miR-146b was found to target IRF5 [112]. To assess the therapeutic potential of miR-146b, *IL-10* KO mice given mimics intraperitoneally (IP) (dose: 10 mg/kg, twice a week) showed improved disease outcome including weight and tissue morphology along with significantly lower IL-12p40- and MHCII-expressing macrophages when compared to scrambled control [112]. miR-146b was further explored as a therapeutic entity by Deng et al. [113]. Their study found the upregulation of miR-146b in mucosal samples three days after 3% DSS withdrawal and detected an abundance in lamina propria cells compared to epithelial cells. Simultaneously, M2-associated cytokines also increased three days after 3% DSS withdrawal [113]. Next, the team loaded miR-146b mimics into mannose-modified trimethyl chitosan (MTC) nanoparticles that could target the mannose receptor on macrophages and delivered them orally (dose: 20 µg/kg) in 3% DSS-treated mice. They observed that, while the colon tissue morphology was damaged to the same extent in both MTC-non-coding and MTC-miR-146b groups, the latter recovered faster in both body weight and epithelial structure three days after DSS withdrawal, along with an increase in M2 phenotype signature proteins in the colonic mucosa [113]. While this established MTC-miR-146b’s effect on macrophage polarization, the group further investigated mucosal regeneration in an in vitro system of miR-146b-treated bone marrow-derived M1 cells atop a scratched “wounded” fetal human colon cell monolayer [113]. The results showed a closure of the wound, mediated by STAT3-dependent IL-10 production from miR-146b-mimic treated macrophages, demonstrating the utility of miR-146b as an anti-inflammatory target as well as bringing about mucosal regeneration [113]. miR-98-5p has also been implicated in macrophage polarization by targeting Trib1, which controls macrophage polarization [114]. DSS (4%)-treated mice (8 weeks old) treated with antagomir-98-5p administered through the caudal vein showed downregulation of M1-phenotype-related cytokines and the upregulation of M2-phenotype-related cytokines resulting in an improved disease activity index [114]. Another strategy of miR-98-5p inhibition was explored using lncMEG3, which targets and inhibits miR-98-5p expression [115]. Upon administration of pcDNA3.1-MEG3 complexed with Lipofectamine 2000 to TNBS (150 mg/kg of TNBS in 50% ethanol, 50% *v*/*v*)-induced Sprague Dawley rats, the MEG3-expressing rats showed reduced expression of miR-98-5p, ROS, TNFα, IL-1β along with higher expression of IL-10 and improved disease activity index [115]. Another miRNA that regulates macrophage polarization is miR-125b, which is enriched in macrophages over lymphoid cells [156]. RAW264.7 macrophages treated with miR-125b were overactivated when stimulated with IFN-γ due to increased expression of IFN-γR on their surface [156]. These cells also displayed higher CD80 expression on their surface, indicating an inflammatory phenotype [156]. Anti-miR-125b-treated cells were less responsive to IFN-γ treatment and exhibited reduced levels of CD80 and reduced T cell activation, suggesting a treatment strategy for IBD [156,157]. 

STAT3 plays a differential role depending on the immune system milieu in which it is activated. In colitis, while STAT3 activation leads to pathogenic T cell survival in the adaptive compartment, it contributes to the suppression of inflammation in colonic epithelial cells and macrophages (reviewed in [158]). STAT3 was shown to be upregulated in the colon of pediatric UC patients. Subsequent investigation revealed that miR-124 targets STAT3 in NCM460 colonic epithelial cells [159], RAW264.7 macrophages, and THP1 macrophages [160]. Consistent with this, STAT3 upregulation correlates with miR-124 downregulation in the colonic biopsies of pediatric UC patients [159] and in the intestinal macrophages of pediatric intestinal failure (IF) patients [160]. A similar correlation between miR-124 and STAT3 expression levels was observed in DSS [159] and *IL-10* KO [159] mouse models of colitis. Moreover, the transfection of miR-124 mimics inhibited the release of pro-inflammatory cytokines upon stimulation of RAW264.7 macrophages [160,161] and THP1 macrophages [160] with bacterial endotoxin lipopolysaccharide (LPS) STAT3 inhibition [160]. Recently, in a randomized controlled Phase 2a trial, Vermeire et al. evaluated the efficacy of orally administered small molecule ABX464 or placebo for eight weeks in 32 patients with moderate to severe UC [162]. The proposed mechanism of action for ABX464 involves the splicing of lncRNA 599-205 that releases miR-124 in macrophages [163] to promote anti-inflammation [164]. While 20 patients received ABX464, 9 patients received placebo. Overall, ABX464 was well tolerated with no severe adverse effects, strongly recommending a Phase 2b trial to start detecting an efficacy signal. 

On the contrary, the downregulation of STAT3 expression was associated with an increase in the transcript levels of pro-inflammatory cytokines. DSS treatment of Caco-2 and HT-29 cells significantly downregulated STAT3 and miR-135a expression, while there was a simultaneous increase in pro-inflammatory cytokine expression [165]. miR-135a showed anti-inflammatory properties, as treatment with mimics activated STAT3 signaling and lowered pro-inflammatory cytokines [165]. Together, these data suggest that the pro-inflammatory or anti-inflammatory effect of STAT3 activation depends on the cellular context.

miRNAs play a role in regulating the plasticity of T cells [166]. miR-10a was upregulated in naturally occurring Treg cells and preserved the Treg phenotype by targeting (B Cell Lymphoma 6) Bcl-6 [166]. miR-10a in IEC lamina propria dendritic cells is controlled by microbiota as seen by differential miR-10a expression in the colon of normally housed mice vs. germ-free housed mice, aiding to maintain intestinal homeostasis [167]. In a clinical context, miR-10a was downregulated in inflamed mucosal tissues vs. normal tissues of UC and CD patients [168]. These authors further studied the role of miR-10a in these patients by lentiviral pre-miR-10a transfection of peripheral blood CD4+ T cells isolated from UC and CD patients, and healthy controls, and observed a reduction in cytokines associated with Th1 and Th17, indicating miR-10a’s role in regulating T cell differentiation [168]. Additionally, miR-10a targeted multiple cytokines implicated in IBD, such as IL-12/IL-23p40, NOD2, and NFκB pathways [167,168]. These investigations reveal the potential of miR-10a mimics in restoring intestinal homeostasis by influencing host microbiota responses. 

miR-155 inhibition in LPS-stimulated RAW 264.7 cells leads to the downregulation of inflammatory cytokines by downregulating pNFκB and NLRP3-related proteins [169]. Along with its role in innate immunity, it also plays a role in adaptive immunity by contributing to the differentiation of CD4+ T cells into various subsets of helper T cells (Th1, Th2, Th17) and Treg cells [170]. Th17 and Treg cells maintain a balance to ensure homeostasis, and dysregulation increases IBD susceptibility [171]. Singh et al. found that 8–12-week-old miR-155 KO mice have lower levels of Th17 cells in the mesenteric lymph nodes in response to chronic DSS-induced colitis (1% DSS administered for seven days followed by seven days of water, treatment regime repeated for three cycles) along with reduced pro-inflammatory cytokines [116]. Zhu et al. tested the therapeutic potential of miR-155 antagomirs in the context of maintaining the balance between Th17 and Treg cells [117]; in C57BL6/J mice (6–8 weeks-old) treated with 3% DSS, IP administration of miR-155 antagomirs (dose: 80 mg/kg, from the 5th day of the DSS cycle, for three consecutive days) significantly improved the disease activity index compared to untreated controls [117]. Furthermore, the authors found that the inhibition of miR-155 in the antagomir-treated group showed a reduced number of Th17 cells and a decline in IL-17A and IL-6 levels but an increase in Treg cells, IL-10, TGF-B1, in the mesenteric lymph nodes, indicating the maintenance of Th17/Treg cell balance as a strategy to alleviate colitis [117]. 

IL-25 is downregulated in inflamed mucosa of IBD patients and participates in CD4+ T cell differentiation into Th1/Th17 via IL-10 [172]. Shi et al. showed an inverse relationship between IL-25 and miR-31 expression in primary mouse colonic macrophages and epithelial cells, the colon of CD patients, and mouse colitis models (TNBS and *IL-10* KO mice) and further confirmed via target validation studies that miR-31 targets the 3′ UTR of IL-25 mRNA [118]. The team explored the therapeutic potential of miR-31 antagomirs in TNBS and IL10 KO mice by intracolonically administering antimiR-31 (dose: 5 mg/kg) complexed with polyethyleneimine (PEI, 25kDa) 12 h after 3 mg TNBS in 100 µL 50% ethanol injection in female BALB/c mice (6–8 week), and weekly administration from ages 12–20 weeks in IL-10 KO mice [118]. Compared to untreated controls, treatment with antimiR-31 in both colitis models improved intestinal histology and weight recovery, and showed a marked decrease in IFN-γ^+^Th1 and IL-17A^+^ Th17 cell populations, indicating a role in alleviating inflammation via the regulation of Th1/Th17 response [118]. 

Another miRNA implicated in Th17 differentiation is miR-301a [119]. Inflamed mucosa and peripheral blood-derived monocyte cells of UC and CD patients showed elevated levels of miR-301a [119]. Furthermore, CD4+ T-cells isolated from the peripheral blood of IBD patients when transduced with miR-301a lentivirus (LV-miR-301a) showed increased transcript levels of IL17a, RORC, and TNFα. In contrast, treatment with LV-anti-miR-301a showed significantly lower levels of these mRNAs compared to controls. Mechanistically, miR-301a targets SNIP1, which is implicated in the suppression of Th17 differentiation. Then, the team tested the effects of miR-301a inhibition by the intracolonic administration of miR-301a in TNBS treated mice, which improved the pathological scores, lowered levels of IL17A, RORγt, TNFα, and increased the expression of Foxp3 in the colonic mucosa when compared to the anti-miR control treatment [119]. 

A recent study explored the role of miR-219a-5p on Th1/Th17 differentiation in IBD [120]. The group studied the inflamed mucosa of CD and UC patients and found the levels of miR-219a-5p to be lower than the normal mucosa of the same IBD patients. Furthermore, isolated peripheral blood (PB) CD4+ T-cells from IBD patients showed lower miRNA levels than PB CD4+ T-cells from healthy donors. Subsequent pro-inflammatory stimulation of PB CD4+ T-cells from healthy donors also revealed low miRNA levels. Additionally, when PB CD4+ T cells from IBD patients were transduced with lentivirus-encoding miR-219a-5p, it reduced IFN-γ+ and IL17A+ CD4+ T cells in addition to pro-inflammatory cytokines. Studying the effects on mice, TNBS-induced colitis mice showed reduced levels of miR-219a-5p in the colorectal tissues compared to healthy controls. Therapeutic intervention by intraperitoneal administration of pre-miR-219a-5p (dose: 5 mg/kg) complexed with PEI for four consecutive days, starting 12 h after TNBS injection, improved the pathological scores compared to control by suppressing Th1/Th17 mediated immune responses [120].

In the CD4 + CD45RO+^hi^ transfer colitis model, authors investigated the effects of antagomiR-142-5p (5 mg/kg) treatment (starting at 5% weight loss for 5 days) by profiling the gene expression in the colons of these mice [58]. Transcriptome analysis revealed a shift in profile towards healthy control mice, presenting a possible therapeutic strategy. Furthermore, the gene sets indicated a correlation with IL10RA activation induced by the inhibition of miR-142-5p, reinforcing its therapeutic utility [58].

While miR-122 inhibition is being evaluated in clinical trials to combat Hepatitis C virus (HCV) infections using the antimir Miravirsen which inhibits miR-122-HCV mRNA interaction [173,174], the miRNA’s role has also been assessed in IBD. In epithelial cells (HT-29), LPS (500 ng/mL LPS, 16 h) induced apoptosis, and NOD2-mediated NFκB activation was repressed by pre-treatment with pre-miR-122 for 2 h, as miR-122 directly targeted the 3′ UTR of NOD2 mRNA [175]. In another study, Selenium-binding protein 1 (SBP1), which is upregulated in inflamed mucosal biopsies of CD patients, was shown to be a direct target of miR-122-5p [121]. Oxidative stress directly affects SBP1 mRNA expression and inversely affects miR-122 expression in HT-29 cells. In a TNBS-induced colitis model (BALB/c mice, 3 mg in 100 µL of 50% ethanol, administered transrectally) treatment with pre-miR-122 (dose: 5 mg/kg, 12 h after TNBS induction, mice killed on day 3) alleviated inflammation as seen by the reduction in pro-inflammatory cytokines, oxidative stress-related proteins and p65NF-κB signaling affected by oxidative stress NFκB. In vitro studies further showed that oxidative stress causes the hypermethylation of the promoter region of miR-122, thereby affecting its expression [121].

The colon of IBD patients has been shown to overexpress Substance P (SP), a neuropeptide that interacts with NK-1R and stimulates miR-31-3p in NK1R-overexpressing colonic epithelial cells (NCM460-NK-1R cells) via the JNK pathway [122]. NCM460-NK-1R cells, when transfected using the siPORT™ NeoFX lipid transfection system with antisense-miR-31-3p, showed an increase in pro-inflammatory mRNAs CCL2 and IL-6. In contrast, miR-31-3p mimics caused a reduction in these cytokine mRNAs in addition to TNFα and IL-1β mRNAs, depicting the miRNAs’ role in negatively regulating pro-inflammatory cytokines in colonic cells. Consequently, inflamed UC colons were shown to express higher levels of miR-31-3p than control samples. Likewise, higher expression was observed in the epithelial layer of colons of 2% DSS (5 days)- and TNBS (intracolonic 80 mg/kg TNBS in 30% ethanol on day 1, colon tissue harvested on day 7)-treated male C57BL/6J mice. Despite the higher colonic levels of miR-31-3p, when LNA-as-miR-31-3p was administered intracolonically (dose: 40 µg, on days 1, 3, and 5 of DSS induced-colitis, and days 2, 4, and 6 of TNBS induced colitis), it exacerbated colitis disease outcome. The group then investigated the effects of overexpression of miR-31-3p in a DSS-induced colitis model by intracolonic administration of agomirs (dose: 80 µg) and found it to alleviate colitis as seen by the reduction in colonic immune cell infiltration, TNFα, CXCL10 and CCL2 mRNA, and improvement in the histopathology score. Target analysis revealed that miR-31-3p targets RhoA, which is implicated in intestinal inflammation, showing that the overexpression of miR-31-3p downregulates RhoA expression, thereby reducing inflammation [122].

Together, these data indicate how mouse models of IBD, in conjunction with patient-derived data and samples, have helped nominate several miRNAs for therapeutic use, with some candidates (e.g., ABX464) progressing to clinical trials.

## 5. Exosome-Aided Communication

Exosomes are extracellular vesicles (EVs) originating from the endosomal system and are 50–150 nm in diameter [176]. These nanovesicles are secreted by cells in healthy and pathological conditions; consequently, they can be of diagnostic utility as they depict the state of the cells from which they originate. It is also increasingly being recognized that EVs can be leveraged as vehicles to transport fragile payloads such as nucleic acids, as they can be relatively easily manipulated either after isolation or endogenously during their genesis. Moreover, EVs are stable, offering the advantage of being well tolerated in the host [177,178]. Exosomes are enriched in ncRNA, especially miRNA, as well as mRNAs, proteins, and lipids, bringing about gene regulation [179]. Indeed, exosomes provide important regulatory signals to recipient cells. Since the gut is home to many cell types and the microbiota, EVs provide a means to maintain intestinal homeostasis [179]. In IBD patients, EVs from the intestinal lumen are abundant in pro-inflammatory mRNAs, and the stability of the payload indicates potential utility as biomarkers to assess intestinal mucosal health [180]. Here, we summarize some recent studies that delved into or utilized miRNA-loaded EVs in cell culture experiments and in animal models of colitis.

Endogenous manipulation of EVs during their biogenesis enables the enrichment of a particular miRNA. As discussed above, miR-146a mimics can be utilized in IBD for their anti-inflammatory properties. Wu et al. produced EVs by transfecting bone marrow-derived stem cells (BMSCs) with recombinant lentivirus, containing miR-146a plasmid and growing these cells until passage 5 [181]. Later, the culture supernatants were purified, and the isolated EVs characterized for miR-146a enhancement, morphology, and surface epitome proteins [181]. The authors tested the miR-146a-EVs in a TNBS Sprague Dawley rat model by injecting 100 µg per rat of the EVs resuspended in 1 mL PBS through the tail vein on the third day of TNBS enema [181]. Colonic analysis of miR-146a expression revealed higher expression following miR-146a-EV treatment, compared to control EVs, and improved the disease activity index. Furthermore, there was a reduction in the levels of pro-inflammatory cytokines and an improvement in the tissue morphology. Validated miR-146a targets in NFκB signaling—TRAF6 and IRAK1—were suppressed following miR-146a-EV treatment, further bolstering the utility of miR-146a-EVs [181].

Serum exosomes are being explored for diagnostic utility since, compared to tissue biopsies, they are easy to collect from patients. In addition, their payloads are expected to lend novel insight into disease pathogenesis and mechanisms and tracking their biodistribution could facilitate their use as delivery vehicles. A recent study explored serum exosomes from CD patients (CD-exo) for their role in aggravating intestinal inflammation and barrier disruption [182]. CD patients had a significantly higher number of exosomal RNAs than healthy volunteers (Ctrl-exo). Upon isolation, when these serum-derived CD-exo were labeled with the fluorescent dye DiR (1,1-dioctadecyl-3,3,3′3′-tetramethylindotricarbocyanine-iodide) and injected IP into mice, they made their way to the intestines. Next, to probe the effects of these CD-exo on inflammation and barrier disruption, the authors treated RAW 264.7 and Caco-2 cells with either CD-exo or Ctrl-exo. They observed that CD-exo-treated RAW 264.7 cells showed the significant upregulation of pro-inflammatory cytokines TNFα and IL-6, compared to Ctrl-exo-treated cells. In Caco-2 monolayers, they observed a loss in barrier function, as shown by the reduced TEER values and an increase in fluorescein isothiocyanate (FITC) dextran (4 kDa) permeability from the apical to the basal side of the monolayers. Furthermore, in DSS-treated mice (2.5% DSS), the IP administration of CD-exo (dose: 200 ug/mouse, every 48 h during DSS colitis model establishment for seven days) worsened colitis compared to both healthy mice and DSS-mice treated with Ctrl-exo, suggesting that CD-exo indeed have a deleterious effect on colitis that could be mediated by their role in propagating excessive inflammation and barrier disruption. To elucidate why CD-exo aggravate colitis, the team investigated the exosomal content from the sera of healthy and mild-severe CD patients using miRNA sequence analysis, which revealed 82 differentially expressed miRNAs between the two groups. Among them, let-7b-5p expression correlated inversely with CD disease severity when the CD group was divided into mild and severe. When transfecting LPS (100 ng/mL, 24 h)-treated RAW 264.7 cells, they observed that let-7b-5p mimics CD-exo (transfected for 48 h via electroporation) and reduces proinflammatory cytokines by dampening the TLR4/NFκB signaling pathway suggesting let-7b-5p’s role in alleviating inflammation via macrophage polarization. Next, the group exploited CD-exo as a vehicle to target the intestine in mice and modified them to contain let7b-5p mimics. Upon IP injection in DSS-treated mice, the let7b-5p mimics loaded in CD-exo significantly relieved colitis, as seen by the improvement in body weight and mucosal tissue damage [182]. Taken together, these data reveal the differential expression of let-7b-5p in serum CD-exo compared to Ctrl-exo, as well as the utility of let-7b-5p mimics in alleviating inflammation. 

## 6. Host–Microbiota Interactions Can Impact IBD Via Modulating miRNA

Dysbiosis, a reduction in microbial diversity, is a hallmark of IBD, underscoring the role of the microbiome in maintaining mucosal homeostasis. Thus, an emerging field of research involves the role of miRNAs in IBD through host–microbiome interactions, and the future is promising with studies aimed at using miRNAs as disease biomarkers and at leveraging probiotics as a treatment. The host–microbiome interactions are bidirectional, where the host immune system can shape the microbiome, and the microbiome affects the host transcriptome [183]. Global microRNA expression profiling of the cecal transcriptome of germ-free and specific pathogen-free Swiss Webster mice revealed at least eighteen differentially expressed miRNAs [184]. These miRNAs are potential gene regulators of the innate and adaptive immune systems, and some have been implicated in the pathogenesis of IBD [185]. In patients with CD, the miRNA profile is differential between affected and non-affected mucosa tissue [186]. Interestingly, communication between the host and the microbiome is enabled through EVs, and the health of the interaction can be assessed by intestinal EVs [187]. A recent study explored how adherent-invasive *E. coli* (AIEC) that atypically colonize the intestinal mucosa of CD patients replicate in cells receiving exosomes from AIEC-infected intestinal cells [188]. The authors found that AIEC-infected intestinal cells secrete exosomes taken up by the surrounding macrophages, triggering an inflammatory response. Upon further analysis, the authors noticed that the exosomes were enriched in miR-130a and miR-30c that target autophagy mRNAs ATG5 and ATG16L1, promoting AIEC to thrive in recipient cells, including macrophages [188]. To test an intervention to promote autophagy by suppressing miR-130a and miR-30c, the group transfected the human adherent colon cancer cell line T84 with either anti-miR-130a and anti-miR-30c together or non-coding (NC) controls, using Lipofectamine 2000 for 24 h, and then infected the cells with AIEC for 12 h. Treatment with anti-miR-130a and anti-miR-30c significantly reduced the levels of miR-130a and miR-30c compared to NC controls. Furthermore, the inhibition of these miRNAs significantly increased the levels of ATG5 and ATG16L1. It also abolished the proliferation of AIEC [188], suggesting that the inhibition of miR-130a and miR-30c in IECs could be a potential therapeutic strategy to restrict AIEC proliferation in CD patients.

## 7. Conclusions

miRNA has been extensively investigated in many diseases for both its diagnostic and therapeutic utility. In both UC and CD, miRNA can help us understand pathogenesis and to develop novel therapeutic strategies. Recent studies have provided insights into how endogenous miRNA and EVs can be manipulated in vitro and in vivo to alleviate disease symptoms. These studies also allow us to appreciate the strong association between miRNA levels and disease phenotypes in various mouse models. Notably, characteristics that define UC and CD are regulated at the miRNA level, including the loss of barrier integrity and immune system dysregulation. Some of these effects can be mediated via EVs, which have been shown to contribute to disease severity. Further defining the intricate networks of miRNA and the genes they regulate will be vital for advancing progress towards interventions that effectively alleviate disease symptoms and maintain remission. Specifically, since miRNA-based combinatorial cancer therapy has shown some promise in oncology, it is likely that a similar combinatorial approach—miRNA therapeutics, together with existing therapeutics, such as aminosalicylates, anti-TNFα inhibitors, corticosteroids, and newer biologic therapies (Vedolizumab and Ustekinumab)—might be beneficial for the prevention of IBD, as well as to avoid recurrence or the loss of response to current therapies.

## Figures and Tables

**Figure 1 cells-10-02204-f001:**
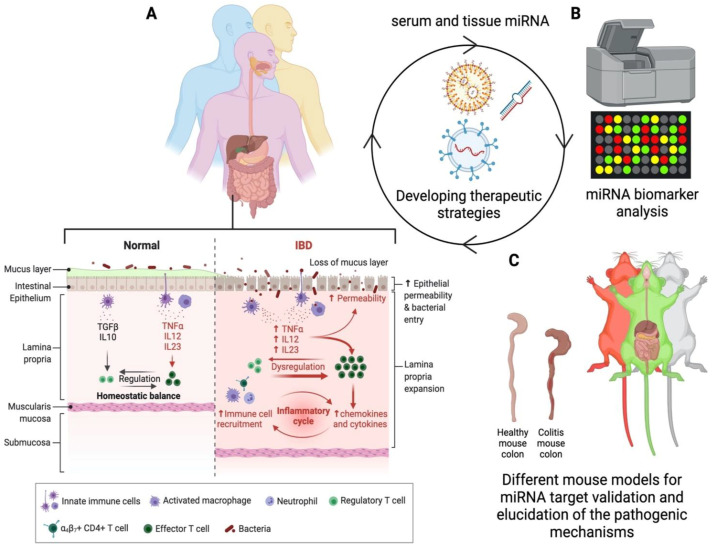
miRNA therapeutics: human–mouse–human translational cycle. (**A**) miRNAs derived from tissue biopsies and serum depict IBD hallmarks, including a dysregulated immune system and disrupted intestinal barrier function. (**B**) miRNAs can be profiled to determine differential levels between IBD patients and healthy volunteers. (**C**) Differentially expressed miRNAs and the complex mechanisms through which miRNAs regulate gene expression can be studied and validated in appropriate murine colitis models, including humanized mouse models and highly genetically diverse mouse strains, which are excellent resources for recapitulating human genetic diversity to elucidate the therapeutic potential of miRNAs. Based on preclinical animal data, therapeutic strategies can be developed, utilizing natural or synthetic nanoparticles.

**Table 1 cells-10-02204-t001:** Mouse models of IBD and their similarities with human disease.

Group	Subgroup	Model Features	Similarities with Human IBD	References
Chemical agents	Dextran Sodium Sulfate (DSS) Colitis Model.	Administration: DSS in different concentrations is dissolved in drinking water; depending upon the dosing conditions, a mouse can develop acute or chronic colitis Pathophysiology: induction of indirect intestinal inflammation due to epithelial injury limited to the gut mucosa—disruption of the epithelial monolayer lining causes the entry of gut microbiome and antigens followed by immune cell activation in the underlying tissues. Disease manifestations: weight loss, bloody diarrhea, colon shortening, crypt abscesses are formed; symptoms in the large intestines are more pronounced than in the small intestines Immune cells: lesions infiltrated by immune cells such as neutrophils and macrophages near the damaged segment; switches from Th1-Th17-mediated acute inflammation to Th2-mediated in chronic colitis exhibiting Th1/Th2 cytokine profile changes Cytokine and Chemokines: increased TNFα, IL-6, IL-17, and Keratinocyte Chemoattractant (KC) in the acute state; increase in IL-4, IL-10, and concomitant decrease in TNFα, IL-6, IL-17; increase in *myeloperoxidase activity.*	Chronic colitis induced by DSS resembles the clinical course of human U—features such as cryptitis and crypt abscesses due to transepithelial neutrophil migration. miRNA commonly dysregulated: miR-223 (↨) *, miR-21 (↨), miR-31 (↑), miR-146a (↑), miR-155 (↑), miR-196a (↓), miR-142-5p (↑), miR-133b (↨), miR-135b (↓), miR-124 (↓).	[43,44,45,46,47,48,49]
Chemical agents	Trinitrobenzenesulfonic acid (TNBS).	Administration: TNBS (haptenating agent) is dissolved in ethanol at different concentrations and delivered rectally; depending upon the dosing conditions, a mouse can develop acute or chronic colitis Pathophysiology: ethanol disrupts the epithelial barrier enabling the interaction with haptenating TNBS, rendering proteins immunogenic, causing acute cell-mediated immune response by acute Th1 inflammation characterized by infiltration of CD4+ T cells. Disease manifestations: dramatic weight loss, inconsistent stool, occult or bloody diarrhea, reduction in colon length, mucosal edema, distorted crypts, formation of abscesses, diffused colonic inflammation Immune cells: T cell tolerance to the microbiota is disrupted, exhibits heightened Th1-Th17 response Cytokines and Chemokines: increased IL-12, IL-23, and IL-17; increased IFNϒ.	The *NOD2* gene is involved in both TNBS-induced colitis as well as human CD. The transmural nature of colitis and edema induced by TNBS is similar to human CD. miRNA commonly dysregulated: miR-223 (↑), miR-196a (↓), miR-10b (↓), miR-21 (↑), miR-133b (↨), miR-135b (↓), miR-142-3p (↑), miR-124 (↓).	[43,47,49,50,51]
Chemical agents	Oxazolone	Administration: intrarectal administration of oxazolone dissolved in ethanol (haptenating agent) Pathophysiology: the haptenating agent induces inflammation in the mucosal layers of mainly the distal colon and is primarily characterized by Th2 cytokine response Disease manifestations: striking drop in body temperature, colon tissue damage, loss of enterocytes and goblet cells, edema, and dense immune cell infiltration Immune cells: higher proportion of CD3+ Ly49c+ and CD4+ natural killer T (NKT) cells, infiltration of neutrophils and monocytes Cytokines and Chemokines: Th2-mediated responses, IL-4, IL-13 (may be induced by IL-33 from damaged epithelial cells or through IL-25), IL-9 (produced because of IL-13 stimulation and TGF-β).	Resembles human UC in terms of intestinal morphology and immunopathogenesis.	[43,52,53]
Spontaneous colitis	SAMP1/YitFcsJ mouse strain	Derivation: selective and continuous brother-sister mating of parental AKR/J mice that developed skin lesions and ileitis formed the SAMP1/Yit colony, which upon constant inbreeding led to SAMP1/YitFc sub strain. Pathophysiology: spontaneous ileitis without genetic, chemical, or immunological manipulation; IFNϒ production at four weeks of age followed by ileitis at ten weeks of age; host-microbial interactions amplify disease severity. Disease manifestations: inflammatory changes are transmural and discontinuous, crypt elongation, villous blunting, infiltration by polymorphonuclear and mononuclear cells in the submucosa, cryptitis, crypt micro abscess formation as the disease progresses. Thickening of the bowel wall characterizes later stages of the disease. Secretory cells outnumber the epithelial/absorptive cells as the disease progresses. Immune cells: Th1 predominant in the early/inductive phase of the disease and changes to a Th1/Th2 chronic phase phenotype as severity progresses. Cytokines and Chemokines: chronic ileitis shows Th1 CD4+ cells that produce higher levels of TNFα and IFNϒ and a higher number of CD8 + α+T-cell receptor T cells.	CD-like ileitis, pathogenesis resembles chronic intestinal inflammation; useful for studying pathways that precede the clinical phenotype. Common features to human CD include disease location, the incidence of extra-intestinal manifestations, perianal disease and fibrostenotic strictures (hallmark features of human CD), and response to conventional therapies.	[54,55]
Spontaneous colitis	C3H/HeJBirLtJ mouse strain	Derivation: selective breeding of the severely affected C3H/HeJ mice with perianal ulcerations resulted in a substrain C3H/HeJBir with a high incidence of perianal ulceration and soft feces. Pathophysiology: mice develop a spontaneous and heritable form of idiopathic IBD, developing defective mucosal immune regulation allowing the activation of pathogenic T cells against gut bacterial antigens Disease manifestations: perianal ulcers occur between 4–7 weeks and once healed, do not recur. Intestinal lesions are present primarily in the cecum, and right colon, characterized by acute and chronic inflammation, ulceration, crypt abscesses, regenerative hyperplasia, and submucosal scarring; less than 10% of cases develop occult blood, soft feces, and large perianal ulcers. Immune cells: CD4+ T cells are quickly activated to produce a Th1-type response, increased IgM B+ cells in Peyer’s patches, increased intestinal S-IgA levels, higher level of activation markers such as CD45RB and CD44 than those from C3H/HeJ, elevated α4β7peripheral blood lymphocytes and in colon intraepithelial lymphocytes when compared to parental levels. Cytokines and Chemokines: CD4+ T cell in response to bacterial antigens produce predominantly IL-2 and IFNϒ.	A valuable resource for genetic and immunological studies of the disease. Mice develop an idiopathic chronic inflammatory bowel illness that is heritable. Acute and chronic inflammation, submucosal scarring, regenerative hyperplasia, crypt abscesses, and ulcerations are found in histopathology investigations of cecum and proximal colon lesions.	[56,57]
Gene knockout (KO)	B6.129P2*-Il10^tm1Cgn^*/J (*IL-10* KO)	Derivation: targeted deletion of IL-10 in mice leads to the development of spontaneous inflammation; deletion of IL-10 from Foxp3+ Treg cells also results in spontaneous colitis. Pathophysiology: immune cell stimulation by gut bacteria is critical for colitis development in this model; the aberrant response of CD4+ Th1-like T cells. Disease manifestations: inflammatory infiltration by lymphocytes, macrophages, and neutrophils. Immune cells: initial inflammation is pro-inflammatory Th1 T cell-driven, progressive increase in Th2 cytokines; excessive inflammatory response may also be due to loss of p100δ induction. Cytokines and Chemokines: after increasing Th1 T cell response (IL-12, IL-17, and IFNϒ), there is a reduction in its production and increased Th2 cytokines IL-4 and IL-13.	IL-10 locus polymorphism has been shown to increase the risk for development of both UC and CD. Colon histopathology shows similarities to human IBD samples. miRNAs commonly dysregulated: miR-223 (↑), miR-101 (↑), miR-155 (↑), miR-31 (↑), miR-142-5p (↑), miR-21 (↑), miR-146a (↑).	[43,49,58,59]
Reconstitution of immunodeficient mice with CD4+ T-cells	Adoptive transfer colitis	Derivation: pancolitis and small bowel inflammation is induced 5–8 weeks after adoptive transfer of naïve CD4 + CD45RB (high) T cells from healthy wild type mice into syngeneic mice (such as RAG KO) that lack T and B cells. Pathophysiology: disruption of T cell homeostasis; good model for examining the early onset of disease inflammation and perpetuation; naïve T cells are susceptible to gut antigens, activating colitogenic T cells secreting cytokines leading to small and large intestinal inflammation. Disease manifestations: transmural inflammation, dense immune infiltration by neutrophils, crypt abscesses; varied degree of weight loss, diarrhea, and loose stools depending on the donor and recipient mouse strain.	This model shows both colitis and small bowel inflammation, which is similar to Crohn’s disease. Disease manifestations, including epithelial cell hyperplasia and cell erosion, transmural inflammation, significant infiltration of leukocytes, and crypt abscesses are similar to those seen in human IBD. Delineates the immunological mechanisms in the induction and regulation of chronic inflammation. miRNAs commonly dysregulated: miR-223(↑), miR-146a (↑), miR-146b (↑), miR-142-5p (↑), miR-21 (↑), miR-203 (↓).	[43,58,60,61]

* (↑), upregulated in both human and mouse; (↓), downregulated in both human and mouse; (↨), inconsistent results between human and mouse.

**Table 2 cells-10-02204-t002:** miRNA tested in relevant preclinical IBD models for improvement of barrier function and reduction in inflammation.

Outcome	miRNA	Model	Treatment	Dose and Regimen	Ref
Occludin upregulation	miR-122a	Intestinal Perfusion Model	Anti-miR-122a	Dose: 25 nM, complexed with Lipofectamine.	[97]
	miR-200c-3p	3% DSS	Antagomir-200c-3p	Dose: 800 mg/day, starting two days before DSS treatment and continued for seven days of DSS course; oral gavage.	[101]
	miR-21	miR-21 KO mice, 3.5% DSS	miR-21 deletion showed less susceptibility to DSS induced colitis	N/A	[102]
Claudin CLDN1 upregulation	miR-29a and b	miR-29 KO mice, TNBS	Mice tolerated TNBS induced barrier disruption	N/A	[103]
Claudin CLDN8 upregulation	miR-223	TNBS	Antagomir-223	Dose: 7.5 mg/kg, prepared as 3 mg/mL in PBS, dosed for three successive days 24 h after TNBS administration; IP administration.	[104]
Claudin CLDN11 upregulation	miR-146b-5p	TNBS	antagomir-146b-5p	Not specified	[105]
Trefoil factor family 3 (TFF3) upregulation	miR-7	TNBS	Antagomir-7	Dose: 100 nmol/kg, tail vein injection, 2 h after TNBS perfusion.	[106]
Bcl-2 upregulation	miR-16	3% DSS	Anti-miR-16	Dose: 5 mg/kg IP administration, twice a week, for the two weeks of 3% DSS administration.	[107]
Epithelial Regeneration by WNT and Hippo signaling	miR-31	3.5% DSS	oxidized konjac glucomannan (OKGM)-PS-miR-31 microspheres	Dose: 3.15 µg, enema. Preventative mode: once per day for 7 d, 2 d gap, 5 d DSS treatment. Therapeutic mode: 5 d DSS treatment, once per day for 7 d; microsphere enema.	[108]
NF-kB pathway dampening	miR-146a	3% DSS	miR-146a mimics	Dose: 5 mg/kg, IP administration.	[109]
	miR-214	DSS	AntagomiR-214	Dose: 12 mg/kg, 2.5 mg/mL diluted in PBS, 4 doses, every 2 days after DSS treatment regime; intracolonic administration.	[110]
IL10RA activation	miR-142-5p	CD4 + CD45RO + hitransfer	AntagomiR-142-5p	5 mg/kg, IP, 5 consecutive days starting at 5–10% weight loss; IP	[58]
NLRP3 inhibition	miR-223	3% DSS	miR-223 mimic	Dose: 50 µg, nanoparticle emulsion- DOPC, squalene oil, PS-20 and an antioxidant, on days 1 and 3 after DSS.	[111]
Macrophage polarization	miR-146b	IL-10 KO	miR-146b mimics	Dose: 10 mg/kg, twice a week; IP administration	[112]
	miR-146b	3% DSS	miR-146b mimic	Dose: 20 µg/kg, encapsulated in mannose modified trimethyl chitosan nanoparticles; oral administration	[113]
	miR-98-5p	4% DSS	Antagomir-98-5p	Caudal vein administration	[114]
	miR-98-5p	TNBS	pcDNA3.1-MEG3 mediated reduction in miR-98-5p	Injected, complexed with Lipofectamine 2000	[115]
Inhibition of Th1/Th17 mediated inflammatory response	miR-155	miR-155 KO, 1% DSS	Lower levels of Th17 upon DSS induced colitis observed	N/A	[116]
	miR-155	3% DSS	AntagomiR-155	Dose: 80 mg/kg, from the 5th day of the DSS cycle, for three consecutive days, IP administration	[117]
	miR-31	TNBS	Antimir-31	Dose: 5 mg/kg, 12 h after TNBS treatment, complexed with PEI; intracolonic administration	[118]
	miR-31	IL10 KO	Antimir-31	Dose: 5 mg/kg, weekly for IL10 KO mice, complexed with PEI; intracolonic administration	[118]
	miR-301a	TNBS	Anti-miR-301a	Dose: 3 optical density, intracolonic administration daily, starting at the day of TNBS induction till 5 d.	[119]
	miR-219a-5p	TNBS	Pre-miR-219a-5p	Dose: 5 mg/kg, complexed with PEI, 4 consecutive days, 12 h after TNBS administration	[120]
Oxidative stress and SBP1 downregulation	miR-122	TNBS	Pre-miR-122	Dose: 5 mg/kg, 12 h after TNBS treatment	[121]
RhoA reduction	miR-31-3p	2% DSS	AgomiR-31-3p	Dose: 80 µg; days 1, 3 and 5 of DSS treatment, intracolonic administration	[122]

DSS: dextran sodium sulfate; TNBS: 2,4,6-trinitrobenzene sulfonic acid; KO: knock out; IP: intraperitoneal; PEI: polyethyleneimine, SBP1: selenium binding protein 1; DOPC: 1,2-dioleoyl-sn-glycero-3-phosphocholine; PS-20: polysorbate-20.

## Data Availability

Not applicable.

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
