# Peer review of "Role of MicroRNA in Inflammatory Bowel Disease: Clinical Evidence and the Development of Preclinical Animal Models"

_cells, 2021, doi:10.3390/cells10092204_

Round 1
Reviewer 1 Report
A very interesting review about the use of micro RNA in inflammatory bowel diseases, also analising the mouse models already present in medical literature. The paper is coincise and complte, and will in my opinion be eligible to be puyblished after minor revisions:
a section in the introduction or a separate matherial and methods section stating what keywords were used and what databses were searched in order to select the articles included in this review would be a great addition to this paper.
Also the introduction should be expanded making a brief description of inflammatory bowel disease and current therapies available; here an article you could consider: doi: 10.1080/03007995.2020.1786681.
The term inflammatory bowel disease refers to 2 different conditions that have clinical, morphological and biological different characteristics: Crohn’s disease and ulcerative colitis. I think the authors should provide a brief description of both conditions, of their difference and available therapies; It would be interesting to discuss if the difference among these conditions may bring to a different use of miRNA in the future. Authors are encouraged to express their opinion on the argument.
Also in the conclusion, defining IBD as a single condition is wrong, as both UC and CD have different characteristic and etiology. So I would probably address the single conditions.
Also in the same paragraph authors are encouraged to give their thoughts about the potential developments of miRNA in the future.
An interesting part that in my opinion should be expanded is the one talking about combination therapy, using mi RNA with other drugs, as proposed in oncology.
Thank You
Author Response
We sincerely thank the reviewers for their time and for providing insightful comments on our manuscript. We provide a point-by-point response to the reviewers’ comments.
Reviewer 1
A very interesting review about the use of micro-RNA in inflammatory bowel diseases, also analyzing the mouse models already present in medical literature. The paper is concise and complete, and will in my opinion be eligible to be published after minor revisions:
A section in the introduction or a separate material and methods section stating what keywords were used and what databases were searched to select the articles included in this review would be a great addition to this paper. The introduction should be expanded making a brief description of inflammatory bowel disease and current therapies available; here an article you could consider: doi: 10.1080/03007995.2020.1786681. The term inflammatory bowel disease refers to 2 different conditions that have clinical, morphological, and biological different characteristics: Crohn’s disease and ulcerative colitis. I think the authors should provide a brief description of both conditions, of their difference and available therapies; It would be interesting to discuss if the difference among these conditions may bring to a different use of miRNA in the future. Authors are encouraged to express their opinion on the argument.
Response: We thank the reviewer for this excellent suggestion. We have updated the introduction section accordingly. Also, see below:
Introduction: Inflammatory bowel disease (IBD) is a multifactorial progressive disease marked by recurrent chronic inflammation of the gastrointestinal tract, a dysregulated immune system, and dysbiosis [1-4]. IBD comprises two main forms—Crohn’s disease (CD) [5] and ulcerative colitis (UC) [6, 7]. Abdominal pain, chronic diarrhea, weight loss, exhaustion, and anorexia are common symptoms among younger CD patients. Bloody diarrhea is a symptom of chronic CD. Although the entire gastrointestinal tract can be impacted in CD patients, the terminal ileum and colon are the most affected. Preferred therapeutic agents for CD include immunosuppressants (e.g., azathioprine, mercaptopurine, and methotrexate), corticosteroids, anti-TNF therapy (e.g., Adalimumab and Infliximab), monoclonal antibodies (e.g., Vedolizumab and Ustekinumab), and surgery for patients who do not respond to treatment [8, 9]. In UC, only the colonic mucosa is inflamed, and the major symptoms include rectal tenesmus, bleeding, diarrhea, abdominal pain, and fecal incontinence. Preferred therapeutic agents include, 5-Aminosalicylic acid (oral, suppository, or enema), corticosteroids, anti-TNF agents (e.g., Infliximab, Golimumab, Adalimumab), and monoclonal antibodies (e.g., Vedolizumab and Ustekinumab)[8, 10]. In the United States, approximately 3 million adults (>18 years of age) have IBD [11], and there is currently no cure. The global prevalence of IBD is increasing [12, 13], necessitating the development of innovative therapeutic strategies. The mechanisms underpinning IBD pathogenesis are still emerging; nevertheless, a large body of evidence suggests that microRNAs play a major role in IBD pathophysiology [14-16]. Here we review clinically relevant miRNAs that have been validated in various mouse models of IBD.
In the conclusion, defining IBD as a single condition is wrong, as both UC and CD have different characteristic and etiology. So, I would probably address the single conditions. In the same paragraph authors are encouraged to give their thoughts about the potential developments of miRNA in the future. An interesting part that in my opinion should be expanded is the one talking about combination therapy, using miRNA with other drugs, as proposed in oncology.
Response: Again, we thank the reviewer for this useful suggestion. We have updated the conclusion section accordingly. Also, see below:
Conclusion: miRNA has been extensively investigated in many diseases for both its diagnostic and therapeutic utility. In both UC and CD, miRNA can help us understand the pathogenesis and to develop novel therapeutic strategies. Recent studies have provided insights into how endogenous miRNA and EVs can be manipulated in vitro and in vivo to alleviate disease symptoms. These studies also allow us to appreciate the strong association between miRNA levels and disease phenotypes in various mouse models. Notably, characteristics that define UC and CD are regulated at the miRNA level, including the loss of barrier integrity and immune system dysregulation. Some of these effects can be mediated via EVs, which have been shown to contribute to disease severity. Further defining the intricate networks of miRNA and the genes they regulate will be vital for advancing progress towards interventions that effectively alleviate disease symptoms and maintain remission. Specifically, since miRNA-based combinatorial cancer therapy has shown some promise in oncology, it is likely that a similar combinatorial approach—miRNA therapeutics together with existing therapeutics, such as aminosalicylates, anti-TNFα inhibitors, corticosteroids, and newer biologic therapies (Vedolizumab and Ustekinumab), might be beneficial for prevention of IBD, as well as to avoid recurrence or loss of response to current therapies.
Reviewer 2 Report
This review article by Suri et al gives an overview of miRNA involvement in intestinal inflammation with an emphasis on animal models of IBD and potential miRNA therapeutic targets. The authors also discuss the current challenges in the field. Overall, the piece is well written. Here are several suggestions.
- It would be much helpful to the readers if the authors can use a table or figure to summarize the information given in the “MicroRNA as a therapeutic target in IBD” section.
- A quick literature search, e.g. in PubMed, pulls out several recent relevant studies (PMID: 31628427, 30779922, etc.) which are not discussed here. The specific miRNAs covered by Suri et al do not overlap completely with those in another recent review (PMID: 34131410). Could the authors clarify their literature search criteria?
- An emerging field of research is how miRNA impacts IBD via modulating host-microbiota interactions. It would be great if the authors could give their perspectives on this area.
- Page 4, line 159: “While the exact etiology is unknown, these four main factors affect specific phenotypes that are characteristic of IBD - immunopathogenic changes, alterations in resident gut microbes, loss of oral tolerance, loss of epithelial barrier function, and polarization of T-cells [27]”. The statement is not clear. The authors say “four main factors” but list five. How is “loss of oral tolerance” involved in IBD?
Author Response
We sincerely thank the reviewers for their time and for providing insightful comments on our manuscript. We provide a point-by-point response to the reviewers’ comments.
Reviewer 2
This review article by Suri et al gives an overview of miRNA involvement in intestinal inflammation with an emphasis on animal models of IBD and potential miRNA therapeutic targets. The authors also discuss the current challenges in the field. Overall, the piece is well written. Here are several suggestions.
It would be much helpful to the readers if the authors can use a table or figure to summarize the information given in the “MicroRNA as a therapeutic target in IBD” section.
Response: This is an excellent suggestion. We have now included a table listing the clinically relevant miRNAs validated in mouse models of IBD (Table 2).
A quick literature search, e.g., in PubMed, pulls out several recent relevant studies (PMID: 31628427, 30779922, etc.) which are not discussed here. The specific miRNAs covered by Suri et al do not overlap completely with those in another recent review (PMID: 34131410). Could the authors clarify their literature search criteria?
Response: We have evaluated studies in which miRNAs identified in human IBDs were validated in mouse models of IBD. Hence, some of our studies do not overlap with current review articles.
An emerging field of research is how miRNA impacts IBD via modulating host-microbiota interactions. It would be great if the authors could give their perspectives on this area.
Response: This is a great suggestion. We have now added a new section on the Host-microbiome in IBD.
- Host-microbiota interactions can impact IBD via modulating miRNA
Dysbiosis, a reduction in microbial diversity, is a hallmark of IBD, underscoring the role of the microbiome in maintaining mucosal homeostasis. Thus, an emerging field of research involves the role of miRNAs in IBD through host-microbiome interactions, and the future is promising with studies aimed at using miRNAs as disease biomarkers and at leveraging probiotics as a treatment. The host-microbiome interactions are bidirectional, where the host immune system can shape the microbiome, and the microbiome affects the host transcriptome [185]. Global microRNA expression profiling of the cecal transcriptome of germ-free and specific pathogen-free Swiss Webster mice revealed at least eighteen differentially expressed miRNAs [186]. These miRNAs are potential gene regulators of the innate and adaptive immune systems, and some have been implicated in the pathogenesis of IBD [187]. In patients with CD, the miRNA profile is differential between affected and non-affected mucosa tissue [188]. Interestingly, communication between the host and the microbiome is enabled through EVs, and the health of the interaction can be assessed by intestinal EVs [189]. A recent study explored how adherent-invasive E. coli (AIEC) that atypically colonize the intestinal mucosa of CD patients replicate in cells receiving exosomes from AIEC-infected intestinal cells [190]. The authors found that AIEC-infected intestinal cells secrete exosomes taken up by the surrounding macrophages, triggering an inflammatory response. Upon further analysis, the authors noticed that the exosomes were enriched in miR-130a and miR-30c that target autophagy mRNAs ATG5 and ATG16L1, promoting AIEC to thrive in recipient cells, including macrophages [190]. To test an intervention to promote autophagy by suppressing miR-130a and miR-30c, the group transfected the human adherent colon cancer cell line T84 with either anti-miR-130a and anti-miR-30c together or non-coding (NC) controls, using Lipofectamine 2000 for 24h, and then infected the cells with AIEC for 12h. Treatment with anti-miR-130a and anti-miR-30c significantly reduced the levels of miR-130a and miR-30c compared to NC controls. Furthermore, inhibition of these miRNAs significantly increased the levels of ATG5 and ATG16L1. It also abolished the proliferation of AIEC [190], suggesting that inhibition of miR-130a and miR-30c in IECs could be a potential therapeutic strategy to restrict AIEC proliferation in CD patients.
Page 4, line 159: “While the exact etiology is unknown, these four main factors affect specific phenotypes that are characteristic of IBD - immunopathogenic changes, alterations in resident gut microbes, loss of oral tolerance, loss of epithelial barrier function, and polarization of T-cells [27]”. The statement is not clear. The authors say “four main factors” but list five. How is “loss of oral tolerance” involved in IBD?
Response: We thank the reviewer for this suggestion. We have now deleted the oral tolerance to list the four major factors.